# Fast, Sensitive and Specific Detection of *Thailand orthohantavirus* and Its Variants Using One-Step Real-Time Reverse-Transcription Polymerase Chain Reaction Assay

**DOI:** 10.3390/v11080718

**Published:** 2019-08-06

**Authors:** Vololoniaina Raharinosy, Jean-Michel Heraud, Harinirina Aina Rabemananjara, Sandra Telfer, Danielle Aurore Doll Rakoto, Claudia Filippone, Jean-Marc Reynes

**Affiliations:** 1Virology Unit, Institut Pasteur de Madagascar, 101 Antananarivo, Madagascar; 2Ecole Doctorale Science de la Vie et de l’Environnement, Faculté des Sciences, Université d’Antananarivo, 101 Antananarivo, Madagascar; 3Plague Unit, Institut Pasteur de Madagascar, 101 Antananarivo, Madagascar; 4Institute of Biological and Environmental Sciences, University of Aberdeen, Aberdeen AB24 3FX, UK; 5Département de Biochimie Fondamentale et Appliquée, Faculté des Sciences, Université d’Antananarivo, 101 Antananarivo, Madagascar; 6Unité de Biologie des Infections Virales Emergentes, Institut Pasteur, 69000 Lyon, France

**Keywords:** Hantavirus, Thailand orthohantavirus, Real-time RT-PCR

## Abstract

Genetic variants of Thailand orthohantavirus (THAIV) have been recently reported from rodents in South-East Asia and in islands from the South-West part of the Indian Ocean. In order to detect THAIV and its variants, we developed a sensitive and specific real-time RT-PCR targeting the S segment. Our assay was developed in two different RT-PCR systems that gave similar results in terms of sensitivity. Moreover, our results demonstrated a specificity of 100%.

## 1. Introduction

Hantaviruses are enveloped RNA viruses belonging to the family *Hantaviridae* (Order *Bunyavirales*) [1]. Hantavirus genome encompasses three negative single-strand segments. Segments Large (L), Medium (M) and Small (S) encode respectively for RNA-dependent RNA polymerase (RdRp), two external glycoproteins (Gn and Gc) and the nucleocapsid (N) protein. Each protein is coded from a gene with a single Open Reading Frame (ORF). However, an additional ORF encoding a non-structural (NSs) protein is added at the S-segment of some hantavirus species carried by certain rodent species [2]. The 3' and 5' terminal sequences of hantavirus genome are more conserved and complementary and can form a hairpin structure. This structure distinguishes viruses belonging to the order *Bunyavirales* [3]. 

Hantaviruses were identified in small mammals as reservoirs hosts (rodents, insectivorous, bats) and more recently in fish and reptiles [4]. Some hantaviruses associated with rodents can spillover and infect humans. Transmission occurs after inhalation of aerosolized infectious particles from the saliva, urine, and feces of infected small mammals [5]. Human infections could present different clinical syndromes from a hemorrhagic fever with renal syndrome (mainly in Eurasia) [6] to cardio-pulmonary syndrome (mainly in the Americas) [7]. Until now, no evidence of human-to-human transmission has been reported except for Andes orthohantavirus [8,9].

In Africa, the first molecular evidence of hantavirus was reported in rodents by Klempa B. et al. [10] by Nested RT-PCR using degenerated primers that target a conserved region of the L fragment of hantaviruses. This method allowed the detection of Anjozorobe virus (ANJZV) in rodents from Madagascar [11,12] and Mayotte virus (MAYOV) in rodents from Mayotte Island [13]. These two viruses that circulate in Indian Ocean islands are genetically closely related and are both genetic variants of Thailand orthohantavirus (THAIV) present in South-East Asia as well as two other variants Serang and Jurong viruses [14]. To date, ANJZV has not yet been reported in humans [11]. We developed a real-time Reverse Transcription-Polymerase Chain Reaction (rtRT-PCR) in order to use a fast and performant solution to detect specifically THAIV and its variants.

## 2. Materials and Methods 

### 2.1. Ethics Approval

This study was conducted according to the institutional ethical guidelines. The Institut Pasteur de Madagascar is guided by the International Guiding Principles for Biomedical Research Involving Animals. This Institution comply with the following laws, regulations, and policies governing the care and use of laboratory animals (Directive 2010/63/EU revising Directive 86/609/EEC on the protection of animals used for scientific purposes was adopted on 22 September 2010; National charter related to Ethics on animal experimentation, Ministry of Education and Research, Ministry of agriculture, livestock and fisheries; Charter of the Institut Pasteur related to Ethics on animal experimentation). Moreover, this institution is approved by the US Office of Laboratory Animal Welfare (Animal Welfare Assurance number F17–00356).

### 2.2. Viral Stock and Tissue Collection

RNA was extracted using QIAamp viral RNA kit (Qiagen, Hilden, Germany) from stocks of hantavirus reference strains available in the laboratory: THAIV (strain 749), Puumala orthohantavirus (strain Montbliart-1-2008), Hantaan orthohantavirus (strain 76–118), and Seoul orthohantavirus (strain Mantenay-Montlin/Rn/FRA/2015/2015.00179). As negative controls, livers from 41 hantavirus-free bred brown rats (*Rattus norvegicus*) were purchased from the Institut Claude Bourgelat (Marcy-l’Etoile, France).

New method of hantavirus detection was compared to the available reference method. Thus, a total of 455 black rats (*Rattus rattus*) were captured in Moramanga district (situated at 98 km East of the capital city Antananarivo) from 2013 to 2016. Organs, including liver and spleen, were collected and stored at –80°C before analysis. Small mammals trapping, and sampling was conducted under authorizations for research from the Madagascar Ministry of Environment and Forests. Organs were disrupted using Tissuelyser II (Qiagen, Hilden, Germany) at 1:10 dilution of culture medium (40% of fetal bovine serum, 1% of Amphotericine, 1% Penicillin-Streptomycin, 1% L-Glutamine and Minimun Essential Medium) with a 5 mm stainless steel beads. After centrifugation, supernatants were recovered and total RNA was extracted using NucleoSpin Dx Virus kit (Macherey-Nagel, Düren, Germany) or QIAamp viral RNA kit (Qiagen, Hilden, Germany) according to the manufacturer’s instructions. Enhanced Green fluorescent protein (EGFP) RNA was added at the lysis phase as an internal control [15].

### 2.3. Design of Primers and Probes

The primers and probe sequences were designed to specifically detect THAIV and its variants by utilizing the Primer3Plus software [16]. Designs were based on published S coding domain sequences (CDS) available in Genbank [17] from THAIV and variant strains as well as representative strains of rodent-borne hantavirus species, tentative species and variants recognized by the International Committee on Taxonomy of Viruses. The alignment of S gene was made using ClustalW of the MEGA 5.2 [18]. Primers and probe were selected in the first 150 nucleotides of S coding region we considered as the most conserved part of the aligned CDS (Appendix A, Table 1).

### 2.4. Amplification of Thailand Orthohantavirus and Its Genetic Variants

The rtRT-PCR was performed by using two different real time RT-PCR platforms. The first thermal cycling was implemented in LightCycler 480 Instrument II (Roche, Meylan, France). Briefly, 4 µL of extracted RNA were added in 16 µL of a reaction mixture containing 0.4 µM of each primer, 0.1 µM of probe, 1× Master Mix, 0.2 µL of mixture of Reverse Transcriptase and hot-start DNA polymerase, 0.4 µL of Rnase inhibitor (SensiFAST Probe No-ROX One-Step kit, Bioline, London, UK). The RT-PCR program consisted in a reverse transcription step at 45 °C for 20 min followed by a denaturation at 95 °C for 10 min and 45 cycles of denaturation at 95 °C for 5 s followed by annealing/extension at 55 °C for 30 s. The second thermal cycling was implemented in Applied Biosystems 7500 Real Time PCR System (Applied-Biosystems, Thermo Fisher Scientific, Waltham, MA USA). In this assay, four µL of extracted RNA were added in 21 µL of a reaction mixture containing 0.4 µM of each primer, 0.25 µM of probe, 1 × Master Mix, 1.5 mM of MgSO_4_, and 0.6 µL of mixture of Reverse/Taq polymerase (SuperScript III Platinum One-Step qRT-PCR System Kit, Invitrogen, Thermo Fisher Scientific, Waltham, MA USA). RT- PCR program consisted in a reverse transcription step at 50 °C for 20 min followed by a denaturation at 95 °C for 10 min and 45 cycles of denaturation at 95 °C for 15 s followed by annealing/extension at 57 °C for 30 s. The published pan-hantavirus nested RT-PCR protocol from Klempa et al. [10] was used in comparison of method and as reference test (Appendix A).

The limit of detection (LOD) of the rtRT-PCR methods was determined using THAIV RNA 10-fold dilutions tested in duplicate (from 3.45E + 05 to 3.45E - 02 FFU/mL of viral stock) and repeated three times.

## 3. Results

### 3.1. Limit of Detection of the Assays

The LOD of the THAIV rtRT-PCR assays were similar for both systems used with the ability to detect up to 3.45 FFU/mL (Table 2). We demonstrated that the LOD of our assay was identical to the one of the pan-hantavirus nested RT-PCR developed by Klempa et al. [10].

### 3.2. Specificity and Sensitivity

The specificity of the assay was evaluated using RNA from different strains of hantavirus available in the laboratory. Amplification curves were only obtained with RNA from the THAIV strain and ANJZV strains (Table 3), whereas, all hantavirus RNAs were detected by the pan-hantavirus nested RT-PCR. On the other hand, we tested the RNA extracted from liver of 41 farmed brown rats known to be uninfected with hantavirus. All these results indicated 100% specificity of the method.

Using our THAIV rtRT-PCR method developed in Applied Biosystems 7500, we tested also RNA extracted from liver and spleen of black rats captured in the wild and previously tested by the. Hantavirus nested RT-PCR reference method [10]. The results showed in Table 4, indicate hantavirus infection regardless of the primers used by the pan-hantavirus (targeting the L gene) or the rt RT-PCR (targeting the S gene). The sensitivity and the specificity were 100% (CI 95%: 96.78–100) and 99.70% (CI 95%: 98.38–99.99) respectively (Youden Index: 0.997).

## 4. Discussion and Conclusions

In this report, we developed a sensitive, specific and robust real-time RT-PCR assay to specifically detect Thailand orthohantaviruses. Our assay was developed in two different amplification systems that gave similar results in terms of sensitivity. Although specificity of our assay in ABI 7500 system could not be fully tested due to unavailability of some strains in Madagascar, we can still consider that specificity of the assay in the two systems was similar since primers and probe used were identical. Since S segment have more conserved region than M or L after sequence alignment, primers and probes used in our assay were designed from the S gene of representative rodent-borne hantaviruses to be specific of the THAIV S gene and its variants (ANJZV). Thus, cross-reactivity with other orthohantavirus outside THAIV is unlikely.

Our assay offers some benefits. Indeed, it can screen rapidly and specifically for the presence of THAIV and its genetic variants. Moreover, the sensitivity was similar to the nested RT-PCR previously developed (Klempa) but with the advantage to avoid potential cross-contamination. We noted that one specimen was positive using our assay but negative using pan-hantavirus nested RT-PCR (Klempa). Due to the limited number of specimens showing this discrepancy (*n* = 1), statistical analyses did not show any significant differences between the two assays. We therefore could not really conclude if this difference can be attributed to a different target for the amplification (S segment versus L segment) or to a slightly greater sensitivity of or assay compared to Klempa one.

This assay can be used for the detection of THAIV and its variants in rodent samples and should be evaluated to detect these viruses from human specimens. Indeed, although period of viremia may limit the use of our assay, it would be interesting to test patients with fever with or without syndrome up to eight days after symptom onset, to address the question of rodent-to-human transmission of THAIV and it is variant. Since higher viral load might be linked to severity of disease with some hantaviruses [20,21], we could implement our assay to test specimens from inpatient presenting renal failure in Madagascar, to explore pathogenicity of ANJZV in human. Serologic tests can also be used to confirm the infection. We are currently developing and in-house antigen based serologic assay for that purpose.

## Figures and Tables

**Table 1 viruses-11-00718-t001:** Primers and probe designed for the detection of the S segment of Thailand orthohantavirus and variants by real-time RT-PCR.

Name	Sequence	Position *
THAIV F S	5’-ATG GCA ACT ATG GAR GAG TTA CAG-3’	46–69
THAIV R	5’-ACA CTY TCM CGG TCA TGI AGT GC-3’	195–172
THAIV P S	[FAM] 5’-CAR CTT GTG GTG GCW AGR CA-3’ [BHQ1]	94–113

* According to the THAIV S sequence GenBank Acc. no AB186420.1 [19].

**Table 2 viruses-11-00718-t002:** Limit of detection of Thailand orthohantavirus (Anjozorobe virus) assay by using the two platforms of real time RT-PCR.

System	LightCycler 480 Instrument II (Roche)	Applied Biosystems 7500 (Applied-Biosystems)	Pan-Hantavirus Nested RT-PCR ^1^
RT-PCR Reagents	SensiFAST Probe One-Step kit (Bioline)	SuperScript III Platinum One-Step Kit (Invitrogen)	
THAIV RNA Concentration (FFU/mL) ^2^	Ct	Ct	
3.45 × 10^5^	21.1	18.7	Positive
3.45 × 10^4^	24.6	22.8	Positive
3.45 × 10^3^	27.8	26.9	Positive
3.45 × 10^2^	31.5	30.8	Positive
3.45 × 10^1^	34.3	32.9	Positive
3.45	36.4	36.2	Positive
3.45 × 10^−1^	Negative	Negative	Negative
3.45 × 10^−2^	Negative	Negative	Negative

^1^ Conventional RT-PCR according Klempa et al. [10]. ^2^ Thailand orthohantavirus reference virus (THAIV strain 749) RNA was extracted from a serial ten-fold dilution of a viral stock at 3.45 × 10^6^ FFU/mL.

**Table 3 viruses-11-00718-t003:** Specificity test of Thailand orthohantavirus (Anjozorobe virus) assay by using two different platforms of real time RT-PCR.

System	LightCycler 480 Instrument II (Roche)	Applied Biosystems 7500 (Applied-Biosystems)
RT-PCR Reagents	SensiFAST Probe One-Step kit (Bioline)	SuperScript III Platinum One-Step Kit (Invitrogen)
**Virus Strain** ^1^		
THAIV	Positive	Positive
HTNV ^2^	Negative	not tested
PUUV	Negative	Negative
SEOV ^2^	Negative	not tested
ANJZV ^3^	Positive	Positive
ANJZV ^4^	Positive	Positive

^1^ THAIV: Thailand orthohantavirus, HTNV: Hantaan orthohantavirus, PUUV: Puumala orthohantavirus, SEOV: Seoul orthohantavirus, ANJZV: Anjozorobe virus. ^2^ HTVN and SEOV strains were not available in Madagascar and then could only be tested at the National Reference Centre for Hantavirus in Lyon (France) using LightCycler 480 Instrument II (Roche, Meylan, France) system. ^3^ Sample with high viral load. ^4^ Sample with low viral load.

**Table 4 viruses-11-00718-t004:** Methods comparison between rtRT-PCR using Applied Biosystems 7500 and reference method nested RT-PCR by Klempa et al. [10].

Applied Biosystems 7500 (Our Assay)	Pan-Hantavirus Nested RT-PCR (Klempa)	
	Positive	Negative
**Positive**	113	1
**Negative**	0	341

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
