# Peer review of "Fast, Sensitive and Specific Detection of Thailand orthohantavirus and Its Variants Using One-Step Real-Time Reverse-Transcription Polymerase Chain Reaction Assay"

_viruses, 2019, doi:10.3390/v11080718_

Round 1

Reviewer 1 Report

The manuscript by Raharinosy et al described the development of real-time PCR assays for detection of Thailand hantaviral specific RNA. Their result showed that the newly developed assay demonstrated perfect results in both sensitivity and specificity. The test would be valuable in the diagnosis study of Thailand infection.

The paper is well written and clear. But still, I have some specific comments and questions:

The introduction should include details of sequence and antigenic variation of hantaviruses to inform the less specialist reader.

There should be a brief commentary discussing the rationale for using target S segment sequences in these assays. Please mention any known cross-reactivity towards other hantaviruses.

In lines, 174-175 authors mention the possible application of the new tests for the screening of human samples. Do they see any limitations using the new test for testing of human samples due to a very short time of viremia of Hantaviruses in humans?

The authors state that the sensitivity and specificity was 100 %, but I see (in table 3) that the one sample that was negative in the reference nested PCR assay was positive in Applied Biosystems assay test. I don't understand this- from the text written before I’m not sure where this one negative in the Klempa assay fits in? Some of this is not very clear. I wonder if the differences appeared because of used S segment sequences as a target DNA differently from Klempa et al., where L segment was used.

Author Response

The introduction should include details of sequence and antigenic variation of Hantaviruses to inform the less specialist reader.

è We replace “nucleoprotein (N) with “Nucleocapsid (N) protein. We also added the following paragraph to the introduction, page 2, line 104-110: “Each protein is coded from a gene with a single Open Reading Frame (ORF). However, an additional ORF encoding a non-structural (NSs) protein is added at the S-segment of some hantavirus species carried by certain rodent species. The 3' and 5' terminal sequences of hantavirus genome are more conserved and complementary and can form a hairpin structure. This structure distinguishes viruses belonging to the order Bunyavirales.”

There should be a brief commentary discussing the rationale for using target S segment sequences in these assays. Please mention any known cross-reactivity towards other Hantaviruses.

è We added the following paragraph to the Discussion and Conclusions, page 6, line 287-290: “Since S segment have more conserved region than M or L after sequence alignment, primers and probes used in our assay were designed from the S gene of representative rodent-borne hantaviruses to be specific of the THAIV S gene and its variants (ANJZV). Thus, cross-reactivity with other orthohantavirus outside THAIV is unlikely.”

In lines, 174-175 authors mention the possible application of the new tests for the screening of human samples. Do they see any limitations using the new test for testing of human samples due to a very short time of viremia of Hantaviruses in humans?

è We agree with author’s comment that the short period of viremia may limit the use of our assay. We still think that it would be interesting to test patients with fever and/or renal syndrome up to 8 days after symptoms onset to address the question of human infected with THAIV and it’s variant. Since higher viral load might be linked to severity of disease with some hantavirus (e.g. Saksida A etal. J Infect Dis. 2008 Mar 1;197(5):681-5. doi: 10.1086/527485. Dobrava virus RNA load in patients who have hemorrhagic fever with renal syndrome.), we could implement our assay in specimen from inpatient presenting renal failure in Madagascar to explore pathogenicity of ANJZV. We have revised this section page 6 line 300-307 accordingly.

The authors state that the sensitivity and specificity was 100 %, but I see (in table 3) that the one sample that was negative in the reference nested PCR assay was positive in Applied Biosystems assay test. I don't understand this from the text written before I’m not sure where this one negative in the Klempa assay fits in? Some of this is not very clear. I wonder if the differences appeared because of used S segment sequences as a target DNA differently from Klempa et al., where L segment was used.

è We apologize for the lack of clarity of this paragraph. In fact, Sensitivity and specificity tests in this study are based on the presence or absence of hantavirus in wild black rats. Thus, amplification of L or S gene indicates a hantavirus infection regardless of the primers used by the pan-hantavirus (targeting the L gene) or the rt RT-PCR (targeting the S gene). The Reviewer is right to mention that one discrepancy between one positive using ABI that appeared negative using Klempa assay. Due to the limited number of specimens showing this discrepancy (n=1), statistical analyses do not show significant differences between the two assays. We therefore could not really conclude if the difference can be attributed to a different target (S vs. L) or to a slightly greater sensitivity of or assay compared to Klempa. We have modified the conclusion and discussion section accordingly (page 6; line 293-298).

Reviewer 2 Report

Raharinosy et al. present development and testing of a one-step real-time RT-PCR assay for Thailand orthohantavirus.  The manuscript is straightforward. The authors test for specificity and sensitivity.  The primers and probes should be presented in the text and not in the supplemental data.  The authors should also clearly state and/or explore if the RT-PCR assay is specific to genome or total viral RNA.  The purpose of the assay appears to be for detection, so differentiation of the vRNA species may not be relevant to the authors, but depending upon how other readers plan to use the assay, the specificity/intent of the should be clarified.  The authors should also clearly state the number of assay replicates performed to generate the Ct values in Table 1.  Since the assay also detects ANJZV, the authors should modify the title since the assay is not specific to THAIV.  Table 3 shows the comparison between the Pan-hantavirus assay on the samples used to develop the THAIV assay.  A secondary confirmation of the virus type using a different methodology such as THAIV specific IHC, sequencing, etc. would strongly strengthen the results suggesting that the THAIV assay RT-PCR assay is specific.

Author Response

Raharinosy et al. present development and testing of a one-step real-time RT-PCR assay for Thailand orthohantavirus. The manuscript is straightforward. The authors test for specificity and sensitivity. The primers and probes should be presented in the text and not in the supplemental data.

è Sequences of primers and probe have been added as Table 1 (page 3).

The authors should also clearly state and/or explore if the RT-PCR assay is specific to genome or total viral RNA. The purpose of the assay appears to be for detection, so differentiation of the vRNA species may not be relevant to the authors, but depending upon how other readers plan to use the assay, the specificity/intent of the should be clarified.

è We thank the reviewer for this comment. We have added a sentence in Discussion and Conclusion Page 6 Line 287-290: “Since S segment have more conserved region than M or L after sequence alignment, primers and probes used in our assay were designed from the S gene of representative rodent-borne hantaviruses to be specific of the THAIV S gene and its variants (ANJZV). Thus, cross-reactivity with other orthohantavirus outside THAIV is unlikely”.

The authors should also clearly state the number of assay replicates performed to generate the Ct values in Table 1.

è The rtRT-PCR assay (Table 1 and renamed as Table 2) was performed three times and each dilution of THAIV RNA was tested in duplicate. Then the sentence “ and repeated three times” was added in Page 4, line 223-224.

Since the assay also detects ANJZV, the authors should modify the title since the assay is not specific to THAIV.

è Good comment. In fact since the assay was designed to detect all THAIV variants, we changed title to “Fast, sensitive and specific detection of Thailand orthohantavirus and its variants using one-step real-time Reverse-Transcription Polymerase Chain Reaction assay.

Table 3 shows the comparison between the Pan-hantavirus assay on the samples used to develop the THAIV assay. A secondary confirmation of the virus type using a different methodology such as THAIV specific IHC, sequencing, etc. would strongly strengthen the results suggesting that the THAIV assay RT-PCR assay is specific.

è Table 3 (renamed Table 4) aimed at testing specificity and sensitivity of our assay with Klempa one form specimen collected in the wild (livers and spleen of wild black rats (Rattus rattus)). These specimens were previously tested using the pan-hantavirus RT-PCR (targeting the L gene). Positive and negative specimens were then tested using our rt RT-PCR (targeting the S gene). We agreed with reviewers that a secondary confirmation of all positive specimens would strongly strengthen the results. Nevertheless, in a previous published study from Raharinosy et al. (see ref. 11), we have sequenced more than 100 positive specimens from black rat from Madagascar using RT-PCR from Klempa and all were confirmed as ANJZV. Therefore, we strongly believe that specimens found positive in the current study using Klempa RT-PCR can be strongly considered as positive, and due to limited fund, we decided not to sequence all of them. Moreover, since Klempa is used as a reference specificity and sensitivity of including Youden index are relevant.

Reviewer 3 Report

This concise technical note describes a new qRT-PCR assay detecting Thailand virus and its variants. Although the article is limited in scope and targeted to a very specific group of readers, it contains valuable data and describes a useful assay which can be used as a reference assay for THAIV. I have no major critical remarks and only a few smaller comments which could improve the manuscript.

1. line 53: Please, note that not only the family names but the order names (Bunyavirales) should be italicized, too.

2. lines 55-57: I find the formulation very misleading and suggesting a broad host range of hantaviruses. However, the opposite is true – hantaviruses are very host-specific – in other words, those hantaviruses infecting mammals are not the same ones which are infecting fish. Moreover, the findings from fish and reptiles are based on transcriptomic data only so far and I would be very careful in the statements about infections. Please, reformulate accordingly.

3. lines 129-130: The formulation in brackets seems to contain some mistakes or wrong formatting. I assume that “3.45 105” is actually 3.45 x 105 or 3.45E+05. Analogously the second number is also probably 3.45 x 10-2 or 3.45E-02.

4. Internal control is only briefly mentioned at lines 99-100 and then 146-147. I understand that this is not the main topic of the article but it would be still useful to provide more details. Particularly, I wonder whether this assay was performed as a duplex assay in parallel or whether it was performed separately. If it was performed as duplex assay (as in the quoted reference) then I wonder why the primers and all other details related to the second assay are not mentioned at all. Please, clarify this issue.

Author Response

line 53: Please, note that not only the family names but the order names (Bunyavirales) should be italicized, too.

è Correction was made accordingly Page 2, line 109

lines 55-57: I find the formulation very misleading and suggesting a broad host range of hantaviruses. However, the opposite is true – hantaviruses are very host-specific – in other words, those hantaviruses infecting mammals are not the same ones which are infecting fish. Moreover, the findings from fish and reptiles are based on transcriptomic data only so far and I would be very careful in the statements about infections. Please, reformulate accordingly.

è We have removed the sentence “Hantaviruses are viruses that could infect a variety of hosts such as mammals (rodents, insectivorous and bats) and also fish and reptiles” and replaced it with: “Hantaviruses were identified in small mammals as reservoirs hosts (rodents, insectivorous, bats), and more recently in fish and reptiles.” in Page 2, line 109-110.

lines 129-130: The formulation in brackets seems to contain some mistakes or wrong formatting. I assume that “3.45 105” is actually 3.45 x 105 or 3.45E+05. Analogously the second number is also probably 3.45 x 10-2 or 3.45E-02.

è Thank you for this suggestion. We have used the formatting “3.45E+05” in the text and the Table 2.

Internal control is only briefly mentioned at lines 99-100 and then 146-147. I understand that this is not the main topic of the article but it would be still useful to provide more details. Particularly, I wonder whether this assay was performed as a duplex assay in parallel or whether it was performed separately. If it was performed as duplex assay (as in the quoted reference) then I wonder why the primers and all other details related to the second assay are not mentioned at all. Please, clarify this issue.

è The internal control (EGFP RNA) was used for all rtRT-PCR reactions performed at the National Reference Center laboratory for Hantavirus in Lyon (France) using the LightCycler 480 Instrument II (Roche). Unfortunately, this IC was not available at the laboratory in Madagascar were rtRT-PCR reaction was performed using Applied Biosystems 7500 Real Time PCR System (Applied-Biosystems, Thermo Fisher Scientific). For information regarding assays performed in the LighCycler 480, the IC was carried out separately and in parallel with our rtRT-PCR test. Before the PCR reaction, two separate PCR mixtures were prepared with specific primers for EGFP RNA and for Thailand orthohantavirus.

Round 2

Reviewer 2 Report

The authors present a revised manuscript.  All previous scientific concerns have been addressed.  The authors should review the document for use of singular or plural throughout.  Minor typos were also found.

Introduction:

Phylogeny should be one paragraph. Transmission should be another.

Line 60 'humans' instead of 'human'

Line 71 'humans'

Line 128 '4' needs to be subscript in 'MgSO4'

Tables: Prefer log10, but will go with journal standard for reporting such numbers

Indicate not indicates line 167

Line 194 symptom vs. symptoms

Line 196 hantaviruses instead of hantavirus. What is meant by 'inpatient'  or is 'in patient' meant?

Author Response

We thank the reviewer for his comment and correction. We are addressing point-by-point response to reviewer's comments.

Kindest regards. 

Phylogeny should be one paragraph. Transmission should be another.

Authors: We have considered this comment and actually put together reservoir and transmission in a new paragraph

Line 60 'humans' instead of 'human'

Line 71 'humans'

Line 128 '4' needs to be subscript in 'MgSO4'

Authors: We have revised manuscript accordingly.

Tables: Prefer log10, but will go with journal standard for reporting such numbers

Authors: We prefer our current format for numbers, but like reviewers, we will go with journal standard

Indicate not indicates line 167

Line 194 symptom vs. symptoms

Line 196 hantaviruses instead of hantavirus. What is meant by 'inpatient'  or is 'in patient' meant?

Authors: We have revised manuscript accordingly.